# ComPAIN—Communication of Pain in Patients with Headache

Emma Eicher [1,*], Sabina Räz [1], Pascale Stucki [1], Cinzia Röthlin [1], Miranda Stattmann [1], Bettina Grossenbacher [1], Eileen Neumann [1,2], Heiko Pohl [1], Yvonne Ilg [3], Anke Maatz [2] and Susanne Wegener [1,*]

1   Department of Neurology, University Hospital Zurich, University of Zurich, 8006 Zurich, Switzerland; sabinamaria.raez@usz.ch (S.R.); pascale.stucki@usz.ch (P.S.); cinzia.roethlin@usz.ch (C.R.); miranda.stattmann@usz.ch (M.S.); bettina.grossenbacher@usz.ch (B.G.); eileen.neumann@pukzh.ch (E.N.); heiko.pohl@usz.ch (H.P.)

2   Department of Psychiatry, Psychotherapy and Psychosomatics, Psychiatric Hospital, University of Zurich, 8032 Zurich, Switzerland; anke.maatz@pukzh.ch

3   Department of German Studies, University of Zurich, 8001 Zurich, Switzerland; yvonne.ilg@ds.uzh.ch

\*   Correspondence: emma.eicher@usz.ch (E.E.); susanne.wegener@usz.ch (S.W.)

**Abstract:** Primary headaches are a common debilitating health condition. Proper diagnosis and treatment depend on patients' communication. We wanted to explore differences in pain communication with a special interest in potential sex differences. Patients visiting our headache unit for the first time filled in two different questionnaires (one *before* entering the consultation and one directly *after* finishing the consultation), through which we captured patients' descriptions of their pain, its potential impact on daily lives, the well-being of our patients and the satisfaction with our consultation. We included a total of 35 patients (22 female, 13 male). Women reported experiencing a greater loss of socially active days during the last 3 months because of headaches compared to men. Furthermore, women were more satisfied with our consultation. In addition, we revealed migraineurs characterize their pain differently than stated in the *International Classification of Headache Disorders* (*ICHD-3*) criteria. The adjective "pressing" (drückend) was used significantly more often by migraineurs compared to patients with tension-type headaches. Nevertheless, in the physicians' written report, the characterization more often contained the *ICHD-3* corresponding adjective "pulsating" (pulsierend). Since the typification of headaches and subsequent therapy depends predominantly on the patients' communication, consideration of the individual pain description and further research on headache characterization are indispensable.

**Keywords:** primary headache; pain communication; sex; migraine; impact; satisfaction; ICHD-3



## 1. Introduction

Primary headaches are defined as idiopathic disorders with constant or recurrent pain, in contrast to secondary headaches, which are caused by another underlying condition [1]. A major diagnostic challenge in primary headaches is the absence of radiographic or laboratory biomarkers [2]. The diagnosis is based on patient description and precise history taking. Many sufferers of primary headaches are often inadequately diagnosed and treated. According to a survey in 1992, out of 20'468 US citizens, only 41% of women and 29% of men suffering from migraine had been properly diagnosed by a physician [3]. Hence, in 2004, a detailed guideline was published by the *International Headache Society* intending to standardize headache diagnosis in clinical routine. As a result, the *International Classification of Headache Disorders (ICHD-3)* was established, defining primary headaches as migraine, tension-type headaches (TTH), trigeminal autonomic cephalalgias (TACs), and various other types [4]. The *ICHD-3* distinguishes headache types based on different localization, character, intensity, duration, associated symptoms, and effective treatment. Therefore, patients' communication patterns remain the key to obtaining the diagnostic information required. In 2007, the use of *ICHD-3* criteria was investigated in an urban emergency department setting in New York City [5]. Due to a great variation between patients'

descriptions, the application of the *ICHD-3* criteria and the reaching of a final diagnosis turned out to be very challenging. These data emphasize that language is dynamic and constantly evolving according to the specific social context. Therefore, pain communication needs to be analyzed carefully in order to reach the proper diagnosis.

In a headache clinic, incorporating narrative-based medicine principles can enhance patient care. Actively listening to patients' stories during consultations provides valuable insights into their headache experiences, including triggers, frequency, and impact on daily life [6]. Moreover, narrative-based medicine fosters a collaborative and therapeutic relationship. By encouraging patients to share their concerns and expectations, healthcare providers create a safe space for open dialogue and shared decision-making. This patient-centred approach ensures that treatment plans consider individual values and preferences [6].

### 1.1. Primary Headaches According to ICHD-3

Migraine (ICHD-3: 1.1 and 1.2) is characterized by pulsating, unilateral pain and sensitivity to sensory stimuli such as light, sound, and movement. Further, migraine can occur with an aura (reversible focal neurological symptoms of short duration) with more than 90% of them containing a visual aura component [4]. The pain has a moderate to severe intensity. Migraine has been classified as the second most prevalent concern in the category Years Lived with Disability in the 2016 Global Burden of Disease Study [7]. Women are more often affected by migraine than men in a ratio of 2.8:1 [1]. Before puberty and after menopause, however, this ratio changes, and men are slightly more likely to suffer from migraines [1].

TTH (ICHD-3: 2) on the other hand, is the most common primary headache with mild to moderate intensity and neither associated with nausea nor vomiting. Photo- and phonophobia, nevertheless, can occur [4]. TTH is characterized as a cranial pressing and/or tightening pain [4]. It can manifest itself episodically or chronically, affecting women a little more often than men [8]. In half of those affected, the pain occurs on both sides. In contrast to migraine, TTH is not exacerbated by routine physical activity such as walking or climbing stairs [4]. It is important to note, however, that TTH can occur simultaneously with migraine, which complicates the recognition of these primary headaches.

TACs (ICHD-3: 3) are a group of headaches characterized by unilateral and parasympathetic autonomic features, mainly ipsilateral to the headache [4]. The TACs include cluster headache (CH), paroxysmal hemicrania (PH), short-lasting unilateral neuralgiform headaches with conjunctival injection and tearing (SUNCT), short-lasting unilateral neuralgiform headaches with cranial autonomic features (SUNA) and hemicrania continua (HC) [8].

Less common primary headaches are classified as other primary headache disorders (ICHD-3, 4) and include primary cough headache, primary exercise headache, primary headache associated with sexual activity, primary thunderclap headache, and furthermore [4].

### 1.2. Sex Differences in Pain Perception and Communication

Population-based research consistently reveals that women have a higher prevalence of pain than men [9]. Further, it has been postulated that women are also more *sensitive* to pain; this phenomenon can be attributed to sociocultural, psychological, but also biological factors [10]. Moreover, female sex hormones cause a higher pain sensitivity in women depending on the stage of the menstrual cycle; women were shown to react with a lower level of stress-induced analgesia and therefore reveal a greater sensitivity to noxious stimuli compared to men [10].

These sex differences in pain sensitivity could be related to structural and functional differences in the brain itself. Studies have compared the hemodynamic of the brain during pain stimuli with heat application through functional magnetic resonance imaging (fMRI) [11]. Magnetoencephalography (MEG) additionally revealed differences in the

dynamic pain connectome between men and women [11]. Therefore, it is quite likely that pain will be perceived and communicated differently among the sexes. However, socio-cultural, psychological, and biological factors affecting pain are usually not considered in clinical routine, especially during initial consultations where the focus is rather set on the application of general diagnostic criteria [12].

Consequently, if potential sex differences in pain development, perception, and communication are not investigated we might fail to offer sex-sensitive diagnosis and treatment [13].

This project is part of the prospective study "ComPAIN–Communication of pain in patients with headache" and seeks to contribute to the field of sex differences in pain communication. We wanted to capture the description of pain in patients with headaches to derive characteristic communication features whilst considering differences between women and men. To achieve this, we gave patients greater freedom to use their own words when asked to characterize pain than in the given *ICHD-3* criteria. We aimed to systematically study pain communication in patients with primary headaches to improve the diagnosis of headache disorders and to better meet these patients' needs, which in turn may lead to greater therapeutic efficacy as well as patients' satisfaction with treatment at our headache unit.

## 2. Materials and Methods

This explorative, mixed-methods research study was conducted at the Department of Neurology of the University Hospital Zurich (USZ) from August 2021 and reports data acquired until November 2022. The study was performed in accordance with the ethics protocol BASEC 2021-00695, approved by the Zurich cantonal ethics committee. Patients scheduled for a first consultation at our tertiary headache department were screened according to the referral letter from their primary care physician. We selected all patients who were likely to suffer from a primary headache disorder. Further inclusion criteria were: Age > 18 years, ability to give informed consent, and capability of understanding and speaking Swiss or High German fluently. Patients suited to participate were informed about the study in an invitation letter and asked in person for their agreement before entering the consultation. Upon agreement, patients had to fill in the items on the first questionnaire (Q1) before entering the consultation. One of the tasks in Q1 offered the opportunity to describe their pain in free text form as can be seen in Figure 1. Hence, we wanted to assess the pain-characterizing expressions patients used before the initial interaction with the physician. To do so, we assessed the number of words used for pain description and evaluated if there were any differences in the extent of pain characterization by taking a closer look at the adjectives within the texts in questionnaire Q1. Nouns and verbs were also included in the analysis in the form of the corresponding adjective if the linguistic meaning or origin was identical. Since we hypothesized that a difference in pain description is mainly determined by headache, we grouped the cohort of patients into those diagnosed with migraine and those with other diagnoses (TTH, mixed headache type, TACs, other) and compared both groups. Additionally, we analyzed whether the adjectives used for pain characterization in the physicians' final reports matched the adjectives used by the patients in the free texts before entering the consultation.

Next, we analyzed whether sex-related differences in the identification with primary headache pain were present. To do so, we compared the use of a first-person narrative in the patients' free texts. First-person narratives represent pain descriptions where the patient places herself/himself as a subject of the text with the pain being described as a part of oneself. In contrast, pain can also be described in a third-person narrative, not being related to one's person.

Furthermore, Q1 assessed the possible impact of headaches on one's daily living. Patients had to count the number of days in the last three months, in which they had been restricted in daily activities due to pain. We asked them for the number of days they had missed work or school, days they had completed less than 50% of work or scholar

tasks, days they couldn't manage the household, days with 50% or fewer household tasks completed, and finally, days in which they had missed out on social events, family time or leisure activities (Appendix A Figure A2). The assessment was performed using an adapted variation of the MIDAS (Migraine Disability Assessment) questionnaire.

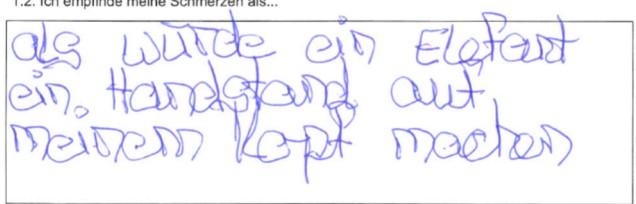

**Figure 1.** Description of pain. Free description of a patient's headache in the questionnaire Q1 (Translation: "As if an elephant was doing a handstand on my head").

Finally, questionnaire Q1 recorded pain localization, temporal development, headache-associated symptoms, pain-reducing or aggravating influences, and patients' well-being (Appendix A Figures A1–A3). We aimed to offer patients different ways to characterize their pain to allow individual preferences and to capture the greater diversity possible in which pain is expressed.

After the consultation at our headache department, patients received a second questionnaire (Q2). In this second questionnaire, participants were asked to rate the interaction and communication during the consultation. Patients evaluated their agreement on nine different statements. Seven of them were phrased positively towards the physician, and two of them were phrased negatively and expressed rather discontent with the consultation (Appendix A Figure A4). Due to interpretation differences of the negatively phrased statements among the participants, we limited the evaluation of the results to the positively phrased statements and categorized them as *satisfied* with the consultation.

### 3. Statistical Analysis

Descriptive statistics were used to describe the whole patient cohort as well as females and males separately. Categorical variables were summarized with frequencies and percentages, and continuous variables with medians and interquartile ranges. Univariate associations between variables and sex were assessed using Chi-squared and Mann-Whitney-U tests for categorical and continuous variables, respectively.

Using linear regression, we further analyzed whether age or headache type revealed an influence on female patients' satisfaction within our consultation. Due to a small study population, we were not able to perform regression analysis on male patients.

The linguistic approach of this study was based on the free texts within Q1. All adjectives used in the texts were collected, grouped, and counted. Nouns and verbs that were meaningfully identical were added to the analysis (for example the noun "pressure" (Druck) was associated with and counted as the adjective "pressing" (drückend)). Associations between headache type and used adjectives for pain description were assessed using Chi-squared tests.

$p$-values $\leq 0.05$ were considered statistically significant. The statistical analysis was conducted with IBM SPSS version 28.0.2.0.

### 4. Results

We included a total of 35 patients, of which 13 were male and 22 female. The distribution of patient data is summarized in Table 1. The median (interquartile range) age was 29 (18–61) years. According to the final clinical reports, 21 (60%) patients were diagnosed with migraine, five (14.29%) with TTH, seven (20%) with mixed headache type (migraine and TTH), one (2.86%) with TACs and one (2.86%) with a not further specified neural-

gia (other). Female patients were younger ($p$ = 0.085) with a median of 25 years (range 18–61 years) versus a median of 39 years (range 19–53 years) and more often diagnosed with migraine (72.73% versus 38.46%, $p$ = 0.030) and mixed headache type (22.73% versus 15.38%, $p$ = 0.030), while male patients more frequently suffered from TTH (30.77% versus 4.55%, $p$ = 0.030).

**Table 1.** Results from questionnaire Q1 and sex comparison.

| | All<br>$n$ = 35 | Male<br>$n$ = 13 | Female<br>$n$ = 22 | *p*-Value |
|---|---|---|---|---|
| **Demographic data** | | | | |
| Age, y (range) | 29 (18–61) | 39 (19–53) | 25 (18–61) | 0.085 |
| **Headache Type** | | | | |
| Migraine, $n$ (%) | 21 (60.00%) | 5 (38.46%) | 16 (72.73%) | |
| TTH, $n$ (%) | 5 (14.29%) | 4 (30.77%) | 1 (04.55%) | |
| Migraine & TTH, $n$ (%) | 7 (20.00%) | 2 (15.38%) | 5 (22.73%) | **0.030** |
| TAC, $n$ (%) | 1 (02.86%) | 1 (07.69%) | 0 (00.00%) | |
| Other, $n$ (%) | 1 (02.86%) | 1 (07.69%) | 0 (00.00%) | |
| **Temporal trend** | | | | |
| Constant pain with light fluctuations, $n$ (%) | 9 (26.47%) | 5 (38.46%) | 4 (19.05%) | |
| (participants) | (34/35) | (13/13) | (21/22) | |
| Constant pain with severe fluctuations, $n$ (%) | 9 (26.47%) | 4 (30.77%) | 5 (23.81%) | |
| (participants) | (34/35) | (13/13) | (21/22) | |
| Pain attacks, in between pain-free, $n$ (%) | 12 (35.29%) | 3 (23.08%) | 9 (42.86%) | |
| (participants) | (34/35) | (13/13) | (21/22) | 0.742 |
| Pain attacks, with pain in between, $n$ (%) | 6 (17.65%) | 2 (15.38%) | 4 (19.05%) | |
| (participants) | (34/35) | (13/13) | (21/22) | |
| **Loss of days/3 months** | | | | |
| No. of workdays [a], $n$ (range) | 3 (0–45) | 0 (0–45) | 3 (0–30) | 0.229 |
| (participants) | (30/35) | (13/13) | (17/22) | |
| Less 50% workdays [a], $n$ (range) | 8 (0–90) | 5 (0–90) | 10 (0–40) | 0.509 |
| (participants) | (30/35) | (13/13) | (17/22) | |
| No. of household days [a], $n$ (range) | 3 (0–67.5) | 0 (0–67.5) | 5 (0–50) | 0.106 |
| (participants) | (31/35) | (13/13) | (18/22) | |
| Less than 50% household days [a], $n$ (range) | 5 (0–90) | 2 (0–90) | 6 (0–35) | 0.535 |
| (participants) | (31/35) | (12/13) | (19/22) | |
| No. of social days [a], $n$ (range) | 5 (0–70) | 4 (0–45) | 7 (0–70) | **0.030** |
| (participants) | (33/35) | (13/13) | (20/22) | |
| **Well-being** | | | | |
| General well-being (points) [b] | 21 (6–29) | 22.5 (6–29) | 21 (6–25) | 0.412 |
| (participants) | (31/35) | (12/13) | (19/22) | |

[a] Counted was the number of lost days in a period of 3 months. [b] The scale of general well-being is set at a minimum of 0 and a maximum of 35 points.

### 4.1. Results from Questionnaire Q1

When we compared the temporal trend of pain over time (Appendix A Figure A1), we found no sex-specific differences (Table 1, temporal trend).

The loss of days at work and for household duties within three months due to primary headaches was similar between males and females. Yet, women reported a significantly higher number of loss of socially active days with 7 days (range 0–70, $p$ = 0.030) versus 4 days loss in men (range 0–45, $p$ = 0.030). We found no significant differences in general well-being and happiness between females and males (Table 1 and Appendix A Figure A3).

### 4.2. Results from Pain Description (Task Q1) and Final Medical Report

To evaluate the individual use of language for pain characterization, we considered how patients described their headaches before the consultation in the form of a freely written text (task 1 in Q1). Results are summarized in Tables 2 and 3. As highlighted in Table 2, we found no relevant differences in the total number of words used for pain description between women and men ($n$ = 12 vs. $n$ = 11, $p$ = 0.870). The range in number of words used by females (3 up to 92 words) and males (1 up to 25 words) was substantial.

**Table 2.** Results from free texts in questionnaire Q1 and sex comparison.

| | All $n = 35$ | Male $n = 13$ | Female $n = 22$ | *p*-Value |
|---|---|---|---|---|
| **Pain Communication** | | | | |
| Number of words for pain description, *n* (range) (participants) | 11 (01–92) (33/35) | 11 (01–25) (13/13) | 12 (03–92) (20/22) | 0.870 |
| First-person narrative, *n* (%) (participants) | 8 (24.24%) (33/35) | 2 (15.38%) (13/13) | 6 (30.00%) (20/22) | 0.431 |

**Table 3.** Results from free texts in questionnaire Q1. Comparison of adjectives used in patients with migraine versus patients with other headaches (no migraine).

| | All $n = 35$ | Migraine $n = 21$ | No Migraine $n = 14$ | *p*-Value |
|---|---|---|---|---|
| **Adjective defining migraine according to ICHD-3** | | | | |
| Pulsating/"pulsierend", *n* (%) (participants) | 9 (27.27%) (33/35) | 4 (20.00%) (20/21) | 5 (38.46%) (13/14) | 0.424 |
| **Adjective defining TTH according to ICHD-3** | | | | |
| Tightening/"beengend", *n* (%) (participants) | 1 (03.03%) (33/35) | 1 (25.00%) (20/21) | 0 (00.00%) (13/14) | 1.000 |
| Pressing/"drückend", *n* (%) (participants) | 18 (54.55%) (33/35) | 14 (70.00%) (20/21) | 4 (30.77%) (13/14) | **0.038** |

Although men used the first-person narrative less often than women, there was no significant difference (15.38% of men versus 30.00% of women, *p* = 0.431) as can be seen in Table 2.

"*Dumpf und pulsierend gleichzeitig. So lange ich mich gar nicht bewege, kein Augenzwinkern, rein gar nichts, und sich auch meine Unterlage (Bett) nicht bewegt, ist es einigermassen auszuhalten. Aber fast schon ein Atemzug von jemand im Zimmer oder einer Fingerrührung meinerseits, kann meinen Kopf bei sehr starker Migräne fast zum Platzen bringen*".

**Citation 1:** Description of pain. Free description of a patient's headache in the Q1. Example of a first-person narrative's use, placing oneself as the subject. This patient was diagnosed with a migraine. (Translation: «Dull and pulsating at the same time. As long as I don't move at all, don't blink my eyes, nothing at all, and my mattress doesn't move either, it is quite bearable. But almost a breath from someone in the room or a finger movement from my part, can almost make my head burst with very severe migraines».)

"*Im Alltag einschränkend. Die Schmerzen sind meist pulsierend mit einem gewissen Druck. Oftmals treten diese mit starken Nacken- und Oberrückenschmerzen auf. Manchmal sind die Schmerzen beidseitig und manchmal einseitig*".

**Citation 2:** Description of pain. Free description of a patient's headache in the questionnaire Q1. Example of third-person narrative and objectification of the pain. This patient was diagnosed with a mixed form of migraine and TTH. (Translation: «Restrictive in everyday life. The pain is usually pulsating with some pressure. It often occurs with severe neck and upper back pain. Sometimes the pain is bilateral and sometimes unilateral».)

According to the *ICHD-3*, migraine is characterized by the adjective *pulsating* (pulsierend). The use of the adjectives *tightening* (beengend) and *pressing* (drückend), however, are predominantly assigned to TTH. Yet, our results differed from the present guidelines (Table 3). We couldn't show a more frequent use of the term *pulsating* (pulsierend)

in migraineurs (20.00% migraineurs versus 38.46% non-migraineurs, $p = 0.424$). We nevertheless found a significant difference regarding the adjective *pressing* (drückend). The migraineurs in our sample used *pressing* (drückend) significantly more often to describe their pain than the patients with other headache types, including TTH (70.00% migraineurs versus 30.77% non-migraineurs, $p = 0.038$).

Finally, we analyzed if the expressions migraineurs used in Q1 before entering the consultation were the same as the adjectives documented for pain description in the final medical reports written by the doctors. Results are shown in Table 4. Seven out of 21 final reports (33.33%) contained the adjective *pulsating* (pulsierend) to characterize headache, although these patients hadn't described them as *pulsating* (pulsierend) before entering the interview. One migraineur out of 21 (4.76%) mentioned *pulsating* (pulsierend) before entering the consultation, yet it was not documented in the corresponding report. Furthermore, three migraineurs (14.29%) used *pulsating* (pulsierend) before entering the interview, followed by subsequent and concordant use of the same adjective in the final report.

**Table 4.** Visualization of the adjectives used by 21 migraineurs before entering the consultation compared to the adjectives used by physicians in the final medical reports.

| Patient ID | Pulsating (Pulsierend) | Constricting (Beengend) | Pressing (Drückend) | Stabbing (Stechend) | Dull (Dumpf) | Dragging (Ziehend) | Burning (Brennend) | % Concordance [a] | Sex |
|---|---|---|---|---|---|---|---|---|---|
| 9 | | | XX | | | | | 100.00 | female |
| 17 | XX | | XX | XX | | | | 100.00 | female |
| 19 | | | XX | | | | | 100.00 | female |
| 20 | XX | | XX | | | | | 100.00 | male |
| 21 | | | XX | XX | | | | 100.00 | female |
| 25 | | | XX | | | | | 100.00 | female |
| 1 | | | X | | | XX | | 50.00 | female |
| 8 | | | XX | X | | | | 50.00 | female |
| 16 | X | | | XX | | | | 50.00 | male |
| 27 | | X | XX | | | | | 50.00 | male |
| 29 | | | XX | | X | | | 50.00 | male |
| 31 | X | | XX | | | | | 50.00 | female |
| 34 | X | | XX | | | | | 50.00 | female |
| 22 | X | X | XX | X | | | XX | 40.00 | female |
| 12 | X | | | X | XX | | | 33.33 | female |
| 28 | XX | | XX | X | | | | 33.33 | female |
| 24 | X | | X | XX | X | | | 25.00 | female |
| 4 [b] | | | X | | | | | 00.00 | female |
| 5 [b] | | | X | | | | | 00.00 | female |
| 11 [b] | X | | | | | | | 00.00 | male |
| 33 [b] | X | | | | | | | 00.00 | female |

X = Adjectives mentioned by the patient before the consultation. X = Adjectives mentioned by the physician in the final report after the consultation. [a] The concordance refers to the percentage of matching adjectives between the ones mentioned by the patient before the consultation and the adjectives written by the physician in the final report after the consultation. [b] The patient either did not fill in the free text or used adjectives that were only mentioned by one single patient (e.g. "disturbing"). To maintain an overview, these adjectives were not integrated into Table 4.

Considering the use of the adjective *pressing* (drückend), three final reports out of 21 (14.29%) contained the adjective *pressing* (drückend) without it being named by the participants before entering the interview. Only one participant (4.76%) described his/her pain to be *pressing* (drückend) without the adjective being found in the corresponding final report. However, we revealed 13 cases (61.90%) in which the use of *pressing* (drückend) by

the participants before the consultation was concordant with the characterization in the physicians' final reports.

In summary, by taking a closer look at the 21 participants diagnosed with migraine, we were able to reveal that the physicians in charge captured the patients' use of the adjective *pressing* (drückend) accurately in 61.90% of the cases. In comparison, however, regarding the adjective *pulsating* (pulsierend), the concordance was only 14.29%. We also found that in 33.33% of patients who had not used the adjective *pulsating* (pulsierend) by themselves before the consultation, their pain nevertheless was being characterized as *pulsating* (pulsierend) in their final reports.

As expected by the higher prevalence of migraine in women, 16 of the 21 migraineurs in our study cohort were women, and five of them were men. Full concordance of all adjectives used for pain characterization was only found in six out of 21 cases ($n = 6/21$, 28.57%), five of them being female ($n = 5/16$, 31.25%) and the sixth participant being male ($n = 1/5$, 20.00%, Table 4). A 50% concordance (half of the adjectives) was found in seven cases ($n = 7/21$, 33.33%) including four women ($n = 4/16$, 25.00%) and three men ($n = 3/5$, 60.00%). 40% concordance was found in one female participant ($n = 1/21$, 04.76%) and 33.33% concordance in two female patients ($n = 2/21$, 09.52%). Finally, one female participant showed a concordance of 25.00% ($n = 1/21$, 04.76%), and no concordance at all was found in four cases ($n = 4/21$, 19.05%) concerning three women ($n = 3/16$, 18.75%) and one man ($n = 1/5$, 20.00%).

### 4.3. Results from Questionnaire Q2

In general, women reported higher satisfaction with the consultation than men (27 points versus 24 points, $p = 0.028$) as can be seen in Table 5.

**Table 5.** Results from questionnaire Q2 and sex comparison.

| | All *n* = 35 | Male *n* = 13 | Female *n* = 22 | *p*-Value |
|---|---|---|---|---|
| Satisfaction with consultation, points [a], (range) (participants) | 26 (10–28) (31/35) | 24 (10–28) (13/13) | 27 (22–28) (18/22) | **0.028** |

[a] The scale of satisfaction with the consultation is set at a minimum of 0 and a maximum of 28 points.

To further evaluate if the sex difference in satisfaction with the consultation was due to different headache types or age, we performed a linear regression (Table 6) adjusted for age and headache type. Age had a high influence on satisfaction ($p < 0.001$), with older patients being more satisfied. Headache type did not have a major effect ($p = 0.191$). The sex difference remains even after adjustment ($p < 0.001$).

**Table 6.** Regression model for the satisfaction with the consultation of female participants.

| Regression Model Satisfaction [a] | Variable | Regression Coefficient | 95% Confidence Interval | *p*-Value |
|---|---|---|---|---|
| | Age | 0.442 | 0.301–0.583 | **<0.001** |
| | Sex | 12.497 | 8.460–16.533 | **<0.001** |
| | Headache | 1.403 | −0.744–3.549 | 0.191 |

[a] This linear regression model is based on the dependent variable being satisfaction with the consultation (minimum = 0 points, maximum = 28 points) and the predictor variables being age, sex, and headache type.

We further performed an explorative analysis to find out if the patients' satisfaction with the consultation was correlated with the percentage of matching adjectives for pain characterization. We hypothesized that a correct reproduction of the patient's pain description in the written report might indicate a better understanding between the patient and physician. However, we didn't find a significant correlation between the patient's satisfaction with the consultation ($p = 0.220$ in the Chi-squared test) and the percentage of correlating adjectives before the consultation versus in the final reports ($p = 0.220$ in the Chi-squared test).

## 5. Discussion

This study aimed to analyze the communication of patients with primary headaches at a tertiary referral centre, and to explore potential sex differences in communicative patterns.

We found that (1) women with headaches showed a greater loss of days with social events, family time, or leisure activities in 3 months compared to men, and that (2) adjectives used by patients with migraine to describe their pain differed from those in the *ICHD-3* classification. Previous studies suggested women with migraine have to deal with a higher burden of disease than men with migraine [7,14]. Therefore, our study confirms previous findings. The reason for this sex difference could be a higher pain sensitivity in women, but pain treatments and their effects were also shown to differ between the two sexes [15].

In addition to the difference in headache impact, we found that female patients were overall more satisfied with the consultation than male patients. Through linear regression, we were able to detect an influence of age on women's satisfaction with the consultation. A higher age resulted in higher satisfaction. Our results are therefore comparable with previous studies on patients' satisfaction. Older age has been found to be a well-known sociodemographic factor to reach higher satisfaction with medical consultations, as has been shown in a meta-analysis of 221 independent studies in 1990 [16].

Like previous studies, we also found the female subpopulation in our study to be more often diagnosed with migraine and mixed headache type (migraine and TTH) than men. Contradictory to the current knowledge and epidemiology, TTH was more often found in male patients [1]. The reasons for this male preponderance in our study cohort are unclear, however, due to the small sample size it should be interpreted with caution.

In addition to comparing sex differences in headache patients (patient-dependent factors), we were also interested in pain communication in different headache types (disease-dependent factors). Since migraineurs represented the largest part of our study population, we compared them to all non-migraine headache types (TTH, mixed headache type, TACs, and other) in our cohort. One of the most striking findings of this study was the frequent use of the adjective *pressing* (drückend) in migraine patients to characterize their headache, whereas the *ICHD-3* recommended term *pulsating* (pulsierend) was not decisively mentioned more often than in other headache types. Therefore, based on the original pain description of patients before the consultation, the diagnostic criteria of the *ICHD-3* for the differentiation of migraine and TTH did not match the final diagnosis. However, we noticed the use of the adjective *pulsating* (pulsierend) in seven final reports of migraineurs (*n* = 7/21, 33.33%) who hadn't used it to describe their headache before entering the consultation. This could either mean that the patients used different adjectives during the interview than in the questionnaires, that the patient and the physician elaborated on different adjectives together, or that the word got imposed on them by the physician during the interview or in the written report. It could be that this is a confirmation bias in medical decision-making, which means giving greater weight to a preliminary diagnosis while dismissing contradictory evidence, combined with a possible framing effect (the use of semantics favouring a given response) [17] to meet *ICHD-3* criteria and reach a certain diagnosis. Due to the limited sample size, conducting a statistical analysis on description differences within the subpopulation of TTH was not possible at this point. Besides the discussed questionnaires in this study, some patients additionally consented to video- and audiotaping their consultation. These recordings weren't yet transcribed and evaluated for this article but will be included in future studies to further explore this discrepancy.

Moreover, it is important to address the impact of pain intensity on pain perception and communication. Individuals often characterize the same type of pain differently depending on its intensity. Our current study did not specifically investigate this aspect, and the patient number was too small to analyze this aspect. Future research should focus on analyzing pain descriptions across varying intensities in a larger patient cohort. Such investigations would provide valuable insights into the nuanced nature of pain perception and enhance our understanding of effective communication.

## 6. Limitations

Limitations of our study include the single-centre design and the small sample size. Furthermore, our data are not representative of an overall population, since only patients with a consultation at our tertiary headache centre were selected, potentially favouring more severely affected patients. However, the prospective design and collection of questionnaires as well as video and audio data in these patients will allow further comprehensive description of pain communication in the hospital setting.

Moreover, the interpretation of the higher burden of disease in women with headaches must be done with caution: Since we revealed our results through an explorative approach, we didn't record our patients' employment level. This could lead to bias, as women are more often employed in part-time jobs compared to men in Switzerland [17], which could emerge in differing subjective perceptions regarding the impact of headaches on the loss of workdays. Also, social sex differences, such as involvement in care work, were not assessed.

We further did not record whether our patients spoke Swiss German or High German as their first language. This may also have an influence on the selection of words for pain characterization.

Beyond that, disease onset, age at diagnosis, and duration of the headache condition have not been investigated for this study. New media platforms, like social media, have changed language usage and could affect how pain is described across age groups. Younger generations may incorporate digital communication elements into their pain narratives, leading to diverse interpretations.

Finally, since the satisfaction of our patients with the consultation may be influenced by which physician oversaw the consultation, we note that 85.71% of the consultations (30/35) were held by the same female physician. The remaining five consultations were distributed and carried out by either one of two other female physicians or another male physician.

## 7. Conclusions

In this explorative study, we were able to identify headache-specific communication differing from current *ICHD-3* criteria, as well as sex-specific differences in pain impact in patients consulting our headache unit. Further characterization of communication in patients with primary headaches is essential, in order to improve diagnosis and treatment of our patients.

**Author Contributions:** Conceptualization, S.W., Y.I., A.M., H.P. and E.E.; methodology, S.W., Y.I., A.M., H.P. and E.E.; validation, S.W. and E.E.; formal analysis, S.W. and E.E.; investigation, S.W., E.E., S.R., P.S., M.S., E.N. and B.G.; resources, S.W. and M.S.; data curation, E.E., S.R., E.N. and B.G.; writing—original draft preparation, E.E.; writing—review and editing, S.W., Y.I., A.M., H.P., M.S., E.N., S.R., P.S., B.G., C.R. and E.E.; visualization, E.E.; supervision, S.W.; project administration, S.W., E.E. and S.R.; funding acquisition, S.W.; All authors have read and agreed to the published version of the manuscript.

**Funding:** This research received no external funding.

**Institutional Review Board Statement:** The study was conducted in accordance with the Declaration of Helsinki and approved by the Ethics Committee of Zurich (protocol code 2021-00695, date of approval 01.06.2021).

**Informed Consent Statement:** Informed consent was obtained from all subjects involved in the study.

**Data Availability Statement:** The data and questionnaires presented in this study are available on request from the corresponding author. The data are not publicly available due to privacy protection.

**Acknowledgments:** This article was reviewed with the assistance of Lisa Herzog from the Department of Biostatistics at the University of Zurich.

**Conflicts of Interest:** The authors declare no conflict of interest.

**Appendix A**

Welche der Aussagen trifft auf Ihre **Schmerzen** in den letzten 4 Wochen am besten zu? (Bitte nur eine Aussage ankreuzen).

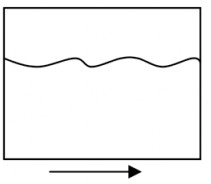 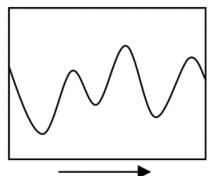 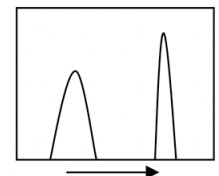 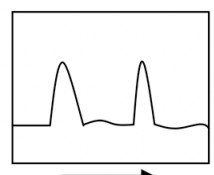

Dauerschmerzen mit leichten Schwankungen | Dauerschmerzen mit starken Schwankungen | Schmerzattacken, dazwischen schmerzfrei | Schmerzattacken, auch dazwischen Schmerzen

**Figure A1.** Item in Q1 about the development of headache pain over time.

Die nächsten Fragen beziehen Sie auf durch Ihre Kopfschmerzen verlorene Zeit.

1. An wie vielen Tagen **in den letzten 3 Monaten** konnten Sie aufgrund Ihrer Kopfschmerzen nicht zur Arbeit oder zur Schule gehen? Bitte geben Sie die Zahl der kompletten Fehltage in den letzten 3 Monaten an:

   \_\_\_\_\_\_\_\_\_\_\_\_\_\_\_\_

2. An wie vielen Tagen **in den letzten 3 Monaten** konnten Sie wegen Ihrer Kopfschmerzen weniger als die Hälfte Ihrer üblichen Schul- oder Arbeitsaufgaben erledigen? Bitte geben Sie die Zahl der Tage an, berücksichtigen Sie dabei **nicht** die Tage aus der 1. Frage, an denen Sie nicht zur Arbeit oder Schule gehen konnten; die Gesamtzahl der Tage aus der Frage 1 und der Frage 2 darf 90 nicht übersteigen:

   \_\_\_\_\_\_\_\_\_\_\_\_\_\_\_\_

3. An wie vielen Tagen **in den letzten 3 Monaten** konnten Sie keine Hausarbeit machen aufgrund Ihrer Kopfschmerzen? Bitte geben Sie die Zahl der Tage an, an denen Sie nichts machen konnten:

   \_\_\_\_\_\_\_\_\_\_\_\_\_\_\_\_

4. An wie vielen Tagen **in den letzten 3 Monaten** konnten Sie aufgrund ihrer Kopfschmerzen weniger als die Hälfte Ihrer üblicherweise erledigten Hausarbeit machen? Bitte geben Sie die Zahl der Tage an, berücksichtigen Sie dabei **nicht** die Tage aus der 3. Frage, an denen Sie gar keine Hausarbeit machen könnten; die Gesamtzahl der Tage aus der Frage 3 und der Frage 4 darf 90 nicht übersteigen:

   \_\_\_\_\_\_\_\_\_\_\_\_\_\_\_\_

5. An wie vielen Tagen **in den letzten 3 Monaten** fielen Sie für Familien-, Sozial- und Freizeitaktivitäten aus aufgrund Ihrer Kopfschmerzen? Bitte geben Sie die Zahl der Tage an:

   \_\_\_\_\_\_\_\_\_\_\_\_\_\_\_\_

**Figure A2.** Item in Q1 about the impact of headache on the patients' living. This task is an adapted variation of the MIDAS (Migraine Disability Assessment) questionnaire.

Bitte schätzen Sie ihr **derzeitiges allgemeines Wohlbefinden** ein. Geben Sie bitte an, wie Sie sich in den letzten 14 Tagen meistens gefühlt haben. Kreuzen Sie dazu auf der 6-stufigen Skala jeweils die Zahl an, die am ehesten auf Sie zutrifft: 0 = trifft gar nicht zu, 5 = trifft vollkommen zu. Bearbeiten Sie alle Aussagen.

| Trotz der Schmerzen würde ich sagen: | trifft gar nicht zu 0 | 1 | 2 | 3 | 4 | trifft voll-kommen zu 5 |
|---|---|---|---|---|---|---|
| 1. Ich habe meine alltäglichen Anforderungen im Griff gehabt. | O | O | O | O | O | O |
| 2. Ich bin innerlich erfüllt gewesen. | O | O | O | O | O | O |
| 3. Ich habe mich behaglich gefühlt. | O | O | O | O | O | O |
| 4. Ich habe mein Leben genießen können. | O | O | O | O | O | O |
| 5. Ich bin mit meiner Arbeitsleistung zufrieden gewesen. | O | O | O | O | O | O |
| 6. Ich war mit meinem körperlichen Zustand einverstanden. | O | O | O | O | O | O |
| 7. Ich habe mich richtig freuen können. | O | O | O | O | O | O |

**Figure A3.** Item in Q1 about the overall well-being of patients.

Bitte füllen Sie den nachfolgenden Fragebogen aus. Wenn Sie die Zahl 0 ankreuzen, trifft die Aussage Ihrer Meinung nach <u>gar nicht</u> zu. Wenn Sie die Zahl 4 ankreuzen, trifft die Aussage für Sie <u>voll und ganz</u> zu.

| | 0 | 1 | 2 | 3 | 4 |
|---|---|---|---|---|---|
| Mein Arzt/meine Ärztin spricht mit mir über meine persönlichen Ziele und Gedanken über die Behandlung. | | | | | |
| Mein Arzt / meine Ärztin und ich sind offen zueinander. | | | | | |
| Mein Arzt/meine Ärztin und ich arbeiten in gegenseitigem Einverständnis auf die Ziele hin. | | | | | |
| Mein Arzt/meine Ärztin und ich haben ein Einverständnis über die Art der Veränderungen, die gut für mich wären. | | | | | |
| Mein Arzt/meine Ärztin scheint mich zu mögen, unabhängig davon was ich tue oder sage. | | | | | |
| Wir stimmen darin überein, woran für mich wichtig ist zu arbeiten. | | | | | |
| Ich glaube, dass mein Arzt/meine Ärztin versteht, was meine Erfahrungen für mich bedeutet haben. | | | | | |
| Mein Arzt/meine Ärztin ist streng mit mir, wenn ich über Dinge spreche, die mir in meiner Situation wichtig sind. | | | | | |
| Mein Arzt/meine Ärztin ist ungeduldig mit mir. | | | | | |

**Figure A4.** Item in Q2 to assess the satisfaction of the patients after the consultation. The first 7 statements are phrased positively (a high ranking equals great satisfaction), whereas the last 2 statements are phrased negatively.

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
