# Peer review of "ComPAIN—Communication of Pain in Patients with Headache"

_ctn, doi:10.3390/ctn7020014_

Round 1
Reviewer 1 Report
Narrative neurology is increasingly recognized. This article is clear and brings important issues for diagnosis and patients education. I would just add 1-2 references on narrative neurology.
Reviewer 2 Report
Very interesting and needed project. I am very impressed by your team overall. I do have a few questions and suggestions
Introduction
- When mentioning TTH and physical activity- TTH is not exacerbated by routine physical activity. I would suggest clarifying this
Methods
- Did you look at age of diagnosis as a variable? I would think that this would influence characterization (possibly patients with onset at a very young age may have more basic descriptions that carry forward through life)
Results
- In 4.2 end of paragraph 1, there may be a better term than "huge"
Overall comments
- Do you think that the descriptions of TTH were influenced by almost all patients being male? I did not see this clearly discussed
- Was there discussion of multiple types of pain? For instance, most of my patients will characterize headache when mild vs severe differently (very often pressure when mild and throbbing when severe). I am unable to read German, so am not able to translate the text in the appendix to understand if this was the case
- This paper overall may be better served to wait until you have the data from the audio and video participants, as there seem to be a few important questions influencing your overall conclusions. For instance, I use a lot of nonverbal data in my consultation (asking people to point, etc) and this may be a source of the difference you see pre- and post-consultation.
Good quality with some very minor editing needed
Round 2
Reviewer 2 Report
Great job! I look forward to future studies from your group
There were only a very few minor typos